# Transcriptome and Functional Comparison of Primary and Immortalized Endothelial Cells of the Human Choroid Plexus at the Blood–Cerebrospinal Fluid Barrier

**DOI:** 10.3390/ijms26041779

**Published:** 2025-02-19

**Authors:** Lea Denzer, Walter Muranyi, Rosanna Herold, Carolin Stump-Guthier, Hiroshi Ishikawa, Carsten Sticht, Horst Schroten, Christian Schwerk, Stefan Weichert

**Affiliations:** 1Pediatric Infectious Diseases, Department of Pediatrics, Medical Faculty Mannheim, Heidelberg University, 68167 Mannheim, Germany; lea.denzer@medma.uni-heidelberg.de (L.D.); walter.muranyi@medma.uni-heidelberg.de (W.M.); rosanna.herold@medma.uni-heidelberg.de (R.H.); carolin.stump-guthier@medma.uni-heidelberg.de (C.S.-G.); horst.schroten@medma.uni-heidelberg.de (H.S.); christian.schwerk@medma.uni-heidelberg.de (C.S.); 2European Center for Angioscience, Medical Faculty Mannheim, Heidelberg University, 68167 Mannheim, Germany; 3Laboratory of Clinical Regenerative Medicine, Department of Neurosurgery, Faculty of Medicine, University of Tsukuba, 1-1-1 Tennodai, Tsukuba 305-8575, Ibaraki, Japan; ishi-hiro.crm@md.tsukuba.ac.jp; 4Core Facility Next Generation Sequencing, Medical Faculty Mannheim, Heidelberg University, 68167 Mannheim, Germany; carsten.sticht@medma.uni-heidelberg.de

**Keywords:** blood–cerebrospinal fluid barrier, choroid plexus, endothelium, transcriptome, Wnt signaling

## Abstract

The human choroid plexus (CP) is the location of the blood–cerebrospinal fluid (CSF) barrier (BCSFB). Whereas the epithelial cells of the CP mainly contribute to the formation of the BCSFB, the vessels of the CP are built by fenestrated endothelial cells. Still, the CP endothelium can contribute to barrier function. By ectopic expression of human telomerase reverse transcriptase (hTERT) in primary human CP endothelial cells (HCPEnCs), we recently generated and characterized immortalized HCPEnCs (iHCPEnCs). Here, we compared primary cells of the sixth passage (HCPEnCs p6) with a lower (p20) and a higher passage (p50) of iHCPEnCs by transcriptome analysis. A high concordance of HCPEnCs and both passages of iHCPEnCs was observed, as only small proportions of the transcripts examined were significantly altered. Differentially expressed genes (DEGs) were identified and assigned to potentially affected biological processes by gene set enrichment analysis (GSEA). Various components of the endothelial barrier-relevant Wnt signaling were detected in HCPEnCs and iHCPEnCs. Functional analysis of HCPEnCs and iHCPEnCs showed equal marginal activation of Wnt signaling, supporting the downregulation of β-catenin (CTNNB) signaling in CP endothelial cells, and a contribution to the barrier function by the CP endothelium was retained until passage 100 (p100) of iHCPEnCs. Overall, our data support the suitability of iHCPEnCs as an in vitro model of the CP endothelium over extended passages.

## 1. Introduction

Endothelial cells, which line the innermost layer of blood vessels, can sense shear stress, adjust the diameter and thickness of the vessel wall accordingly through molecular signaling pathways, and control the migration of immune cells and the transport of molecules in and out of the bloodstream [1,2,3]. In addition, they maintain the supply of the underlying tissue and the homeostasis of the microclimate of the different tissues. They also support the immune system in defending against pathogens, tumors, and exogenous material [4,5]. To meet the different demands of the various surrounding tissues and microenvironments, different types of endothelial cells have evolved, which can be classified as organism-wide or organ-specific [6]. The group of organism-wide endothelial cells includes arterial, venous, capillary, and lymphatic cells, whereas organ-specific endothelial cells include endothelia of the heart, liver, lung, kidney, and the blood–brain barrier (BBB), which represents one of the barriers between the central nervous system (CNS) and the remainder of the body [6]. These distinct types of endothelia are distinguished on the basis of their characteristic structure and function, facilitating their organ specificity [7,8,9]. Accordingly, there is a classification into continuous, fenestrated, or discontinuous (also sinusoidal) endothelia, which contribute to various degrees to the regulation of the permeability of blood vessels for water and dissolved nutrients [10,11].

A fundamental mechanism within embryogenesis and tissue homeostasis is the so-called Wnt signal transduction, which is initiated by the secretion of glycoproteins from the Wnt protein family [12,13]. Wnt signaling regulates fundamental aspects of cell development, differentiation, proliferation, and survival, and can utilize diverse signaling pathways and receptors. As a result, it has been suggested that endothelial cells are particularly sensitive to Wnt signaling and may also stimulate themselves through this pathway. Loss- and gain-of-function experiments with parts of the Wnt signaling pathway lead to marked changes in vascular development and endothelial cell specification [14]. Activation of the Wnt signaling pathway can be observed in many different vessel types, including the brain microvasculature, during vasculogenesis and angiogenesis in the course of embryonic development [14,15].

Wnt signaling is differentiated into the canonical and non-canonical pathways [13]. The canonical signaling pathway is characterized by the activation of the cytosolic signaling molecule β-catenin (CTNNB), a transcriptional co-activator that, in the absence of a stimulus by a Wnt protein, is constantly degraded. The stabilization of CTNNB following the activation of Wnt signaling favors its accumulation in the cytosol and, eventually, its translocation to the nucleus, where CTNNB interacts with members of the DNA-bound T cell factor/lymphoid enhancer factor (TCF/LEF) protein family. This abolishes the repression of Wnt target genes and allows their expression [12,13,16,17].

The CNS requires a strictly homeostatic and sterile environment to maintain its functionality. By shielding the CNS from the rest of the body, both metabolite influx and the efflux of metabolic end products can be controlled. One of the barriers that enable these functions is the BBB, which, in conjunction with astrocytes and pericytes, consists of a specialized continuous endothelium presenting dense tight junctions and low endocytic activity [18]. Importantly, the preservation of barrier properties in mature CNS vasculature is regulated by Wnt signaling [19].

The inner blood–cerebrospinal fluid barrier (BCSFB), located at the choroid plexus (CP) in the ventricular system, represents an additional barrier between the CNS and the blood [18]. In contrast to the BBB, barrier function at the BCSFB is attributed to the epithelium, and the endothelium is characterized by fenestrae with diaphragms containing plasmalemma vesicle-associated protein (PLVAP). However, recent observations point to an important contribution of the endothelium to the barrier function at the CP that can involve the regulation of Wnt signaling [20,21].

Recently, by ectopic expression of human telomerase reverse transcriptase (hTERT) in primary human CP endothelial cells (HCPEnCs), our laboratory has generated and characterized immortalized HCPEnCs (iHCPEnCs) that exhibit typical characteristics of the primary CP endothelium, such as the formation of PLVAP-containing caveolae and fenestrae [21]. Initial analysis of iHCPEnCs confirmed high similarity to primary CP endothelial cells up to passage 30 and revealed differences when compared to other endothelial cell types, such as human brain microvascular endothelial cells (HBMECs) and human umbilical vein endothelial cells (HUVECs) [21]. To investigate whether iHCPEnCs stably represent the CP endothelium, primary cells of the sixth passage (HCPEnCs p6) were compared to iHCPEnCs of a lower (p20) and a higher (p50) passage by massive analysis of cDNA ends (MACE). Using GSEA, identified differentially expressed genes (DEGs) were assigned to biological processes potentially affected by the altered expression of DEGs. The expression of various components of the endothelial barrier-relevant Wnt signaling was analyzed in HCPEnCs as well as iHCPEnCs, and the activation of Wnt signaling was investigated. Finally, the contribution to the barrier function at the CP by iHCPEnCs up to passage 100 (p100) was determined in a two-cell-type in vitro model.

## 2. Results

### 2.1. Transcriptome Analysis of Primary and Immortalized Human CP Endothelial Cells

Immortalized cell lines, as the iHCPEnCs recently described by our laboratory [21], are, unlike cultures derived from primary cells, not subject to natural senescence and, thus, can be used for multiple purposes in research. Still, they should resemble the primary cells as much as possible. A measure to determine the similarity between primary cells and a derived cell line is their respective transcriptomes. To determine whether the immortalized cell line iHCPEnCs exhibited stable gene expression across advanced passages, two different passages were analyzed by MACE. A lower passage of primary HCPEnCs (p6; “HCPEnCs p6”), as well as a low passage of immortalized iHCPEnCs (p20; “iHCPEnCs p20”) and a higher passage of iHCPEnCs (p50; “iHCPEnCs p50”) were examined. Three biological replicates of the primary cells and of each of the two different passages of iHCPEnCs were included in the analysis.

To compare the determined transcriptome data of the three cell types, a heat map of all biological replicates was considered first (Figure 1). As 0.94 is the lowest value, the Pearson coefficient proves a strong positive correlation between the samples. HCPEnCs p6, iHCPEnCs p20, and iHCPEnCs p50 transcriptomes cluster by cell type and passage number, and HCPEnCs p6 are more similar to iHCPEnCs p20 than to iHCPEnCs p50. Overall, however, a high degree of correspondence between the primary cells and the immortalized cells of the two passages can be demonstrated.

### 2.2. Determination of Differentially Expressed Genes (DEGs) and Verification by qRT-PCR

Despite the high degree of correspondence between the primary and the two passages of immortalized cells, it is of significant interest to investigate their differences. Therefore, lists of differentially expressed genes (DEGs) were generated to statistically assess the differences in transcriptomes between primary and immortalized cells, as well as between low and high passages. Overall, 26,586 genes were identified (Appendix A).

To validate the lists of DEGs determined by MACE, qRT-PCR analysis of selected genes was performed to compare their expression levels to the relative expression levels determined in qRT-PCR. The genes of von Willebrand factor (VWF), tissue-type plasminogen activator (PLAT), thrombospondin1 (THBS1), serpin E1 (serpin family E member; SERPINE1), and endothelin 1 (EDN1) were selected because their expression appeared significantly altered after immortalization compared to the primary cells.

The relative differential expressions determined from qRT-PCR and the fold changes and FDR values derived from MACE are compared in Table 1. In qRT-PCR, genes whose fold change is greater than 2 are defined as overexpressed, and those whose fold change is less than 0.5 are defined as underexpressed, whereas in MACE, fold changes between −1.5 and 1.5 were classified as not altered. Table 1 shows that the applied criteria in MACE and qRT-PCR lead to concordant classifications of genes as up- or downregulated.

### 2.3. Expression Level Analysis of the Primary HCPEnCs and iHCPEnCs of Lower and Higher Passages

The expression levels of the overall 26,586 investigated transcripts in HCPEnCs p6, iHCPEnCs p20, and iHCPEnCs p50 were compared with each other. The resulting comparisons are termed p20_vs_p6, p50_vs_p6, and p50_vs_p20, respectively.

The data were filtered for a log2FoldChange > 1.0 and a *p*-value and FDR < 0.05. Genes in this category were considered significant. This analysis identified a total of 2899 genes for p20_vs_p6, 3038 genes for p50_vs_p6, and 4791 genes for p50_vs_p20, whose expressions were significantly different from each other (Appendix A and Figure 2a). The DEGs detected were further divided into downregulated (Down) and upregulated (Up) genes (Figure 2a). However, it is not clear from this subdivision to what extent these groups differ from each other or to what degree they coincide with each other. To clarify this, various comparisons of the significant DEGs of p20_vs_p6, p50_vs_p6, and p50_vs_p20 were made (Figure 2b). The lists of significant DEGs of p20_vs_p6 and p50_vs_p6 were contrasted, as both were based on the comparison of immortalized iHCPEnCs to primary HCPEnCs, to specify how many of the DEGs occurred with both passages of iHCPEnCs. A total of 1806 genes were determined that were assigned only to p20_vs_p6, and 1945 genes occurred only in p50_vs_p6. The remaining 1093 genes detected in both groups were therefore identified as differentially expressed when comparing the primary cells (p6) with both passages of the immortalized cells (p20 and p50). To estimate to what extent the DEGs found with p20_vs_p6 and p50_vs_p6 are regulated in the same or the opposite direction, this group was then compared with the significant DEGs from p50_vs_p20. A total of 769 genes were determined that were significantly altered only with respect to p6, but they did not appear when comparing p50 to p20. At the same time, 4467 genes were identified that differed between the two immortalized cell lines, but they did not show significance when compared to the primary culture. In addition, 324 genes could be identified that were present in both groups (Figure 2b).

### 2.4. Identification of the Genes with the Highest Significances

To get a better overview of the genes with the highest changes in expression levels, a heat map of the 50 DEGs with the highest significances was created (Figure 3a), which can be divided into several distinct clusters. One cluster includes the genes *FAM102A*, *NFE4*, *PIM3*, and *COL4A1*. These genes are only weakly expressed in the primary HCPEnCs p6, but are strongly expressed in the iHCPEnCs, with a stronger expression in the lower passage, p20, compared to the higher passage, p50. Another cluster consists of the genes *CD44*, melanoma cell adhesion molecule (*MCAM*), and *CDPC1*. While the lowest expression of these genes is present in the samples from iHCPEnCs p20, the highest expression is found in the samples from iHCPEnCs p50. In the primary cell samples, the remaining genes are highly expressed and represented by the last distinct cluster, which is extensive compared to the other two clusters. Considering the samples of iHCPEnCs p20 and p50, this extensive cluster can be further subdivided into three different clusters. The first cluster comprises genes that seem to be not significantly altered in p50_vs_p6 and are downregulated in p20_vs_p6 (*ZSCAN18*, *CXCL2*, *COBLL1*, *UCHL1*, *ACTN1*, *CXCL6*, *PCDHGA8*, *PLAU*, *CORO1C*, *NRG1*, *PODXL*, *CXCL1*, *SERPINEB2*, *C6ORF132*). The second cluster includes genes that are more strongly downregulated in p20_vs_p6 compared to p50_vs_p6 (*LOX*, *BCL2A1*, *CSRP1*, *SOAT1*, *ADAMSTL1*, *CNTN1*, *HEG1*, *PCXM2*, *MPLZ2*, *CELF2*, *MMP1*, *PDCD1LG2*, *CCNND1*, *KCNJ12*, *NFH*, *NNAT*). The third cluster comprises genes that display the lowest expression in iHCPEnCs p50, and seem to be progressively downregulated from p6 to p50 (*COL8A1*, *CCN2*, *MT2A*, *PTX3*, *ANKRD1*, *TSTD1*, *SERPINE1*, *TPM1*, *CCND2*, *INHBA*, *MAP1A*, *MYL9*).

To provide a better indication of the distribution of all DEGs, volcano plots for all genes from the p20_vs_p6, p50_vs_p6, and p50_vs_p20 comparisons were generated (Figure 3b). In contrast to the heat map, the log2 fold change, which is plotted against the log10 of the *p*-values, forms the basis for the representation of the significant DEGs in the volcano plot. Log2 fold changes of +1 or −1 and log10 *p*-values of +0.05 or −0.05 were selected as significance thresholds. According to these criteria, most of the genes are classified in the non-significant range. In addition, some of the genes with the highest significances were identified for each group. When comparing the low passage of the immortalized cells to the primary cells (p20_vs_p6), the majority of significant genes were downregulated, including all genes with the highest significances, with the exception of *LAMB4*. Also, when comparing the late passage of the immortalized cells to the primary cells (p50_vs_p6), most significant genes were downregulated, but *PROCR*, *SNHG25*, *CLDN5*, and *STC1* were upregulated, in addition to *LAMB4*. This shift in differential expression toward upregulation can particularly be seen when comparing iHCPEnCs p50 with iHCPEnCs p20 (p50_vs_p20). In addition to the majority of significantly regulated genes being upregulated, *UCHL1* and *CD44*, which are found among the strongly downregulated genes in p20_vs_p6, were assigned to the strongly upregulated genes in p50_vs_p20. Still, when comparing p20_vs_p6 and p50_vs_p6, the classification of several genes remains unchanged.

### 2.5. Effects of Significantly Expressed Genes on Biological Processes and Pathways

Gene set enrichment analysis (GSEA) was performed for p20_vs_p6, p50_vs_p6, and p50_vs_p20 to determine whether, within a group of genes, certain classes of genes were significantly overexpressed or underexpressed, and thus could contribute to an altered phenotype. As a basis for a first GSEA, gene sets (GSs) were selected based on the Gene Ontology (GO) database and ordered according to the biological processes (BPs) as parameters (Gene Ontology Biological Processes; GOBPs). Additionally, a second GSEA based on the Reactome pathway database and a third GSEA based on the Kyoto Encyclopedia of Genes and Genomes (KEGG) pathways were performed. The 10 GSs with the highest GeneRatios and lowest adjusted *p*-value are shown in the form of dot blots in Figure 4 (GOBPs), Figure 5 (Reactome), and Figure 6 (KEGG).

During the GOBP analysis for p20_vs_p6, most of the 10 considered GSs were found in the domains “development” and “morphogenesis”. In contrast, the Reactome pathway analysis of p20_vs_p6 revealed that six of the identified GSs can be assigned to the umbrella term “cell cycle”. The KEGG pathway analysis revealed mainly GSs belonging to the categories “genetic information processing”, “cellular processes”, and “human diseases”. Interestingly, during the GOBP and the Reactome pathway analyses, all GSs had a negative enrichment score (ES), and their normalized ES (NES) was also negative, which is why these GO termini are considered downregulated or suppressed, whereas the KEGG pathway analysis identified both upregulated and downregulated GSs.

For p50_vs_p6, following GOBP analysis, two GSs of development and one set of morphogenesis were found among the 10 GSs considered, while four sets can be assigned to signaling pathways. In addition, two GSs, of “RNA processing” and “positive regulation of cellular component organization”, were identified. During the Reactome pathway analysis, three GSs were determined that can be summarized with RNA procession and translation. Three others can be assigned to the field of biosynthesis, with two dealing with the synthesis of cholesterol and one set with the synthesis of triglycerides. Two sets of genes deal with the regulation of gene expression via SREBF/SREBP, and one set deals with signal transduction via receptor tyrosine kinases. During the KEGG pathway analysis, most GSs were found to belong to the category “cellular processes”, followed by “environmental information processing”. Whereas all GSs identified during the GOBP and the KEGG pathway analyses were downregulated (negative ES and NES), the Reactome database analysis identified both upregulated and downregulated GSs.

When p50 and p20 are compared directly, without prior selection for an intersection with p6, following GOBP analysis, three of the ten considered GSs can be assigned to immunity, three involve different biosynthetic processes, and three involve cell organization as well as cell division. Following the Reactome pathway database analysis, five GSs were identified, which can be assigned to the areas “cell cycle” and “cell division”. Two other sets include “metabolism of steroids” and “fatty acid metabolism”, one includes “cilium assembly”, and two include “RHO GTPase effectors” and “RHO GTPase activate formins”. The GSs found during KEGG pathway analysis were diverse, with most of them belonging to the category “genetic information processing”. The GSs identified for p50_vs_p20 were found upregulated (positive ES and NES) for the GOPBP and Reactome database analyses, whereas the KEGG pathway analysis identified both upregulated and downregulated GSs.

### 2.6. Lack of Extrinsic Activation of the Wnt Signaling Pathway in HCPEnCs and iHCPEnCs

In the CNS, the Wnt signaling pathway contributes to the regulation of angiogenesis and differentiation of endothelial cells and plays an equally important role in the establishment and maintenance of the BBB and the blood–retinal barrier in the differentiated capillaries [22,23,24,25]. We performed reverse transcribed PCR (RT-PCR) on HCPEnCs and iHCPEnCs to evaluate the expression of Wnt ligands, Wnt receptors and co-receptors, Wnt modulators, and signal transducers and targets (Appendix A). Both HCPEnCs and iHCPEnCs express all components required for a functional canonical Wnt pathway, although we could not detect *FZD3* and *FZD10* in HCPEnCs and iHCPEnCs (Appendix A). However, not all components within the groups of Wnt ligands and Wnt modulators could be detected. Only the Wnt ligands *WNT2B*, *WNT3*, *WNT4*, and *WNT9A* (Appendix A) and the Wnt modulators *SFRP1*, *SFRP3*, *SFRP4*, *DKK1*, and *DKK3* (Appendix A) were expressed in HCPEnCs and iHCPEnCs. Notably, iHCPEnCs displayed the expression of *WNT10B*, in contrast to HCPEnCs (Appendix A).

To analyze functional concurrence between HCPEnCs and extended passages of iHCPEnCs, we used HCPEnCs and iHCPEnCs up to passage 100 (p20, p50, p100). We confirmed the comparable expression of ectopic hTERT in all three passages of iHCPEnCs (Appendix A). To investigate the activation of the canonical Wnt signaling pathway in HCPEnCs and iHCPEnCs, HCPEnCs and iHCPEnCs (p20, p50, p100) were treated with the Wnt agonist WNT3a. The translocation of the normally membrane-bound CTNNB into the cytosol and subsequently into the nucleus was analyzed by immunofluorescence (Figure 7). Additionally, treatment with 10 mM LiCl was carried out. Treatment with LiCl as a positive control for a functioning canonical Wnt signaling pathway has been performed previously [26]. LiCl prevents phosphorylation and inactivation of CTNNB by GSK-3β, thus mimicking a functional Wnt signaling pathway [27]. Finally, a specific inhibitor of GSK-3β (SB216763; 10 µM) was employed that had also been used previously [28]. As can be seen in Figure 7, all three treatments did not increase or only marginally increased the detection of CTNNB in the nucleus.

### 2.7. iHCPEnCs Retain Contribution to Barrier Function at the BCSFB

Human epithelial choroid plexus papilloma (HIBCPP) cells [29] present an in vitro model of the human BCSFB and display a strong barrier function when grown on cell culture filter inserts [30]. We have recently shown that iHCPEnCs can specifically contribute to HIBCPP barrier function in a two-cell-type model [21]. To evaluate whether iHCPEnCs retain the ability to contribute to the barrier function over extended passages, HCPEnCs and iHCPEnCs up to passage 100 (p20, p50, p100) were cultivated together with HIBCPP cells in the co-culture model, and the barrier function was determined by measuring the transepithelial electrical resistance (TEER). As can be seen in Figure 8, co-culture with HCPEnCs and all passages of iHCPEnCs (p20, p50, p100) enhanced the barrier function of HIBCPP to a similar extent.

## 3. Discussion

The CP is a complex, highly vascularized structure in the ventricles of the brain, built of different cell types, including endothelial, epithelial, immune, mesenchymal, glial, and neural cells. In addition to being important during brain development and essential for the production of the cerebrospinal fluid, the CP also forms the inner BCSFB, which plays a central role during the pathogenesis of CNS infections and neurodegenerative diseases. The capillaries of the CP are lined by a fenestrated endothelium, which separates the lumen from the stroma and allows a regulated supply to the underlying tissue. Many functions of the CP result from its ability to separate blood and CSF by a stable barrier [31,32,33,34]. Although the main barrier function of the BCSFB is performed by the dense epithelium of the CP, recent data have indicated that the endothelial cells of the CP can contribute to the barrier function [20,21], indicating a need for including CP endothelial cells during in vitro analyses of the BCSFB. An immortalized human CP endothelial cell line (iHCPEnCs) was recently generated in our laboratory by ectopic expression of hTERT [21]. To investigate whether iHCPEnCs stably represent the CP endothelium, a lower (p20) and a higher (p50) passage of iHCPEnCs were investigated in comparison to the primary cells (p6) to detect possible changes in the immortalized cells depending on progressive passaging.

Particularly relevant for the suitability of the iHCPEnCs as an in vitro model of the CP endothelium is the similarity to the primary cell line as well as the stability of gene expression during passaging. Due to the extensive influence of hTERT on cellular processes and signaling pathways [35], the overexpression of hTERT is expected to have a significant impact on gene expression. Still, a high degree of expression level correspondence between HCPEnCs and iHCPEnCs is reflected in a Pearson coefficient consistently between 0.94 and 0.98 obtained by comparison of HCPEnCs p6, iHCPEnCs p20, and iHCPEnCs p50. Here, the greatest agreement in the levels of the gene expressions examined was present when comparing the lower passage, p20, with the primary cells, while the higher passage, p50, showed the greatest differences from the lower passage, p20. The observations that the primary cells are more similar to iHCPEnCs p20 than to iHCPEnCs p50, and that the level of correspondence between iHCPEnCs p20 and iHCPEnCs p50 is similar to that between IHCPEnCs p20 and the primary cells, point to differences in the transcriptomes of lower and higher passages.

When considering the proportions of significant DEGs and the total number of transcripts examined, only 18% of the analyzed genes are significantly differentially expressed in the comparison p50_vs_p20, which implies a still high similarity of these two passages. Importantly, in both the comparisons p6_vs_p20 and p6_vs_p50, only about 11% of the total 26,586 transcripts examined were differentially expressed. How the changes in gene expression are altered during passaging can be determined superficially by dividing them into up- and downregulated DEGs. It was particularly noticeable that the proportion of downregulated DEGs decreased and the proportion of upregulated DEGs increased during passaging. In part, this could be a matter of cellular corrections that compensate for deviations that occurred previously, e.g., as a result of the immortalization of the cells. This is supported by the fact that some DEGs, which were significantly differentially expressed in p20_vs_p6, were no longer identified in p50_vs_p6. DEGs that were upregulated in p20_vs_p6 and downregulated in p50_vs_p6 or vice versa also support this hypothesis.

By looking at individual DEGs, it is neither possible to draw conclusions nor to predict whether or how these changes affect the cell and what phenotype it will potentially develop as a result. To obtain information about the “biology” of the changes, GSEA of the significantly regulated DEGs was performed to identify biological processes in which the immortalized cell lines differ from the primary cells. When GSEA was performed based on the Reactome pathway database, the GSs with the highest GenRatios and lowest adjusted *p*-values differed between iHCPEnCs p20 and iHCPEnCs p50, again pointing to differences between lower and higher passages. Interestingly, when GSs were selected based on the Gene Ontology (GO) database and ordered according to the biological processes (BPs), the 10 GS with the highest GeneRatios and lowest adjusted *p*-values for the comparisons p20_vs_p6 and p50_vs_p6 contained five identical GSs: “circulatory system development”, “epithelium development”, “anatomical structure formation involved in morphogenesis”, “regulation of cellular component movement”, and “regulation of protein phosphorylation”. In accordance, genes belonging to these GSs were among the top 50 significantly altered genes: *NRG1*, *COL4A1*, *HEG1*, *LOX*, *COL8A1*, *ANKRD1*, *TPM1*, *SERPINE1*, and *CCN2* for “circulatory system development”, *NRG1*, *COL4A1*, *HEG1*, *PODXL*, *CCND1*, *NHBA*, and *SERPINE1* for “epithelium development”, *ACTN1*, *COL4A1*, *CSRP1*, *PODXL*, *COL8A1*, *INHBA*, *ANKRD1*, *TPM1*, *SERPINE1*, and *CCN2* for “anatomical structure formation involved in morphogenesis”, *NRG1*, *ACTN1*, *PLAU*, *PODXL*, *NEFH*, *TPM1*, and *SERPINE1* for “regulation of cellular component movement”, and *NRG1*, *HEG1*, *LOX*, *UCHL1*, *CNTN1*, *CCND1*, *INHBA*, *CCND2*, and *CCN2* for “regulation of protein phosphorylation”. This indicates similar changes in biological processes in both passages p20 and p50 of iHCPEnCs compared to the primary cells, and points toward changes in cellular development. A more detailed investigation of the possible changes in cellular processes in iHCPEnCs compared to HCPEnCs requires further investigation.

The differentiation of endothelial cells is one of the early stages of vascular development. During the course of vascular development, endothelial cells acquire specialized properties that include control of cell permeability, and expression of specific transcellular transport systems as well as membrane adhesion molecules to meet the specific needs of different organs [6]. Likewise, the required properties for the regulation of blood flow, the production and binding of chemokines, and the control of leukocyte transport must be developed. These processes of cellular differentiation, but also other cellular mechanisms such as the cell cycle, apoptosis, and cellular communication, are controlled via the Wnt signaling pathway [36]. Activation of the Wnt signaling pathway can be observed in many different vascular types, including the brain microvasculature, during vasculogenesis and angiogenesis in embryonic development [14,15]. The Wnt signaling pathway also plays a major role in the formation of the CP from the cortical seam, and its activity has been demonstrated in murine and human embryonic CPs [37,38]. On the other hand, low CTNNB signaling has been shown in CP endothelial cells from adult mice [39]. Interestingly, the CPs of the four ventricles in mice seem to differ morphologically, and the molecular causes of these differences are as yet unknown [40]. It is important to determine whether the activity of Wnt signaling in iHCPEnCs is the same as in the primary cells. Using RT-PCR, we verified the expression of all essential components of the canonical Wnt signaling pathway in both HCPEnCs and iHCPEnCs. Stimulation with LiCl, the GSK-3β inhibitor SB21676, and the Wnt agonist WNT3a caused none or only marginal nuclear accumulation of CTNNB in HCPEnCs and iHCPEnCs up to passage 100. This result indicates that Wnt signaling is strongly inhibited in CP endothelial cells. One possibility is that, in addition to Wnt signaling, the activation of additional signaling pathways is required for the nuclear translocation of CTNNB [41,42]. Interestingly, in a mouse model, Wnt signaling was only marginally active at a steady state level but was upregulated by an inflammatory stimulus [20]. Furthermore, it has been shown that an increased export can lead to low levels of CTNNB in the nucleus [43,44,45]. The exact reasons for the low levels of CTNNB in the nucleus after stimulation of Wnt signaling require further investigation, including colocalization analyses with further cellular markers, to obtain a more quantitative impression.

It is important that extended passages of immortalized cells retain the functional properties of their primary counterparts. We have previously shown that iHCPEnCs enhanced the barrier function of HIBCPP cells in a two-cell-type model of the human CP, indicating an important contribution of the CP endothelium to the function of the BCSFB [21]. Here, we confirmed that iHCPEnCs up to passage 100 can support increased TEER values generated by HIBCPP cells, which are comparable to those caused by co-culture with the primary HCPEnCs.

Our data demonstrate a high similarity of extended passages of iHCPEnCs to the primary HCPEnCs on the transcriptome level over an extended range of passaging. Importantly, iHCPEnCs up to passage 100 are comparable to primary HCPEnCs in their response to extrinsic activation of the Wnt signaling pathway, and they retain the ability to enhance the barrier function of CP epithelial cells. In summary, these results underline the suitability of iHCPEnCs as an in vitro model of the CP endothelium. Still, although our data indicate that passages of iHCPEnCs up to 100 retain the functional properties of the primary cells, we suggest working with rather early passages of iHCPEnCs, since differences in the transcriptome of lower and higher passages were observed.

## 4. Materials and Methods

### 4.1. Cell Culture

Primary HCPEnCs and iHCPEnCs [21] were cultured in 25 cm^2^ cell culture flasks with Complete Classic Medium^TM^ (Cell Systems, Kirkland, WA, USA) supplemented with CultureBoost^TM^ (Cell Systems, Kirkland, WA, USA) at 37 °C and 5% CO_2_. For experiments with subsequent extraction of RNA, production of protein lysates, or to determine the cellular localization of CTNNB, cells were seeded in 24-well plates, with coverslips sterilized in 100% ethanol to perform immunofluorescence. Cell culture flasks, wells, and coverslips were coated with Attachment Factor^TM^ (Cell Systems, Kirkland, WA, USA) immediately before cell contact.

hCMEC/D3s [46] were grown in EndoGRO^TM^ endothelial medium supplemented with 5% fetal bovine serum (FBS), 50 µg/mL ascorbic acid, 0.75 U/mL heparin sulfate, 1.0 µg/mL hydrocortisone hemisuccinate, 10 mM L-Glutamine, 5 ng/mL rhEGF, and 0.2% EndoGRO LS-Supplement (Millipore, Darmstadt, Germany).

HIBCPP cells were cultured in Dulbecco’s modified Eagle’s medium (DMEM/F-12) supplemented with 10% FCS, 5 µg/mL insulin, and penicillin/streptomycin (100 U/mL penicillin; 100 mg/mL streptomycin) as described previously [21].

### 4.2. RNA Isolation

For RNA isolation, cells were washed once with DPBS before they were lysed by adding modified RLT buffer (according to the instructions of the RNeasy Micro Kit, Quiagen Hilden, Germany). Lysis was assisted by using a cell scraper (for larger cultures) or by pipetting up and down several times. The RNeasy Micro Kit from Qiagen was used for subsequent isolation and purification of RNA. The quantity and quality of total RNA isolated were determined using the Nanodrop^®^ ND1000 UV spectrophotometer at a wavelength of ʎ = 260 nm and 260/280 nm.

Human fetal brain total RNA was obtained commercially (Takara Bio, Mountain View, CA, USA; cat. no. 636526).

### 4.3. RNA Sequencing via Massive Analysis of cDNA Ends (MACE)

The RNA sequencing (RNA-Seq) method of massive analysis of cDNA ends (MACE) was performed by GenXPro to analyze the transcriptome [47]. While conventional RNA-Seq methods randomly fragment and sequence transcripts, MACE specifically sequences transcript fragments from the 3′ end. This prevents multiple fragments of a transcript from being included in the final data quantification, which would cause the overrepresentation of long fragments [47]. In addition, each transcript is marked with a unique sequence as a barcode (“TrueQuant”-unique molecular identifiers) to prevent PCR bias. Instead, each read of a sequence is assigned to only one transcript in the sample, so that even rare transcripts are detected at a 10- to 20-fold lower sequencing depth compared to conventional RNA-Seq [47]. Accordingly, total RNA from HCPEnCs of passage 6 (p6; “HCPEnCs p6”) and from iHCPEnCs from passages 20 (p20; “iHCPEnCs p20”) and 50 (p50; “iHCPEnCs p50”) were isolated from 3 biological replicates each, and their concentration was determined. A minimum concentration of 10 ng/µL for processing was required.

Rapid MACE-seq was used to prepare 3′ RNA sequencing libraries. Samples of 100 ng of DNA-depleted RNA were used for library preparation, using the Rapid MACE-Seq kit (GenXPro GmbH, Frankfurt, Germany). Fragmented RNA underwent reverse transcription using barcoded oligo(dT) primers containing TrueQuant unique molecular identifiers, followed by template switching. PCR-amplified libraries were purified by solid phase reversible immobilization beads, and subsequent sequencing was performed using the Illumina platform NextSeq 500. Unprocessed sequencing reads were adapter-trimmed and quality-trimmed using Cutadapt (version 3.4 [48]). Deduplication based on UMIs (unique molecular identifiers) was performed using in-house tools. FastQC (0.11.9, https://www.bioinformatics.babraham.ac.uk/projects/fastqc/; accessed on 25 January 2020) was used to assess the quality of sequencing reads. Processed sequencing reads were mapped with a reference genome using Bowtie2 (2.4.4, [49] or nvBowtie from nvbio [50]). Quantification of mapped reads to each gene was performed using HTSeq (version 2.0.2 [51]). Raw counts were geometric-mean-normalized using DESeq2 (version 1.38 [52]) and TPM-normalized by dividing each count by the sum of all counts for each sample multiplied by one million. MultiQC (version 1.12 [53]) was used to create a single report visualizing output from multiple tools across many samples, enabling global trends and biases to be quickly identified.

### 4.4. Statistical Analysis of RNA Sequencing Data: Differentially Expressed Genes and Gene Ontology

After the processing and analysis of the samples by GenXPro, the obtained data were further analyzed bioinformatically. R (version 4.4.2), a widely used programming language, was mainly used for statistics and data analysis. Most of the functions and tools used in this analysis were obtained from Bioconductor (version 3.20), a public source with a collection of program libraries for the processing of biological data. Rstudio (version 2025.04.0) was used to edit the source code.

Differential expression analysis was performed using the limma package, version 3.62.1, in R. A false positive rate of α = 0.05 with FDR correction was taken as the level of significance. Volcano plots and heat maps were created using the ggplot2 package (version 3.5.1) and complexHeatmap (version 2.22.0 [54]). The pathway analysis was performed with the fgsea package, version 1.32.2 [55].

The gene set information was obtained from the geneset package, version 0.2.7, using pathway information from KEGG [56], Reactome [57], and the Gene Ontology Consortium [58].

All MACE data described in this study have been submitted to the Gene Expression Omnibus (http://www.ncbi.nlm.nih.gov/projects/geo/; accessed on 1 October 2023) under accession number GSE244600.

### 4.5. Reverse Transcription PCR (RT-PCR)

For synthesis of cDNA from the isolated RNA of treated and untreated cells, the AffinityScript QRT-PCR cDNA Synthesis Kit from Agilent Technologies (Waldbronn, Germany) was used according to the manufacturer’s instructions. The resulting cDNA was stored at −20 °C for further analysis. PCR was performed with a Taq DNA polymerase kit according to the instructions of the manufacturer (Qiagen). After an initial denaturation (94 °C, 2 min) and 35 cycles of denaturation (94 °C, 30 s), annealing (60 °C, 45 s) and elongation (72 °C, 45 s) were performed, followed by a final elongation step (72 °C, 5 min).

For quantitative RT-PCR (qRT-PCR), the Brilliant II SYBR Green QRT-PCR Master Mix Kit from Agilent Technologies was used according to the manufacturer’s instructions. The qPCR was performed using the Stratagene Mx3005P system and MX software (version 4.10). Following an initial denaturation (95 °C, 15 min), subsequently, 40 cycles of denaturation (94 °C, 15 s), annealing (60 °C, 30 s), and extension (72 °C, 30 s) were run using the 1 plateau pre-melt/RT segment and normal 2-step amplification setting. This was followed by the determination of a dissociation curve. During the PCR reactions, a no-template control was included. The efficiency-corrected 2^−ΔΔCT^ method was applied for the calculation of fold changes. To avoid variations due to reference genes, the CT values and efficiencies of the two reference genes, RPL13a (60S ribosomal protein L13a) and SDHA (succinate dehydrogenase complex flavoprotein subunit A), were averaged and used as reference CT and reference efficiency, respectively. A gene was categorized as having higher or lower expression than in the control condition as soon as the fold change was greater than 2 or less than 0.5, respectively.

PCR primers for qRT-PCR were validated as follows, taking into account the MIQE guidelines [59]. To verify primer specificity and accuracy, the PCR product size was confirmed by agarose gel electrophoresis, and a melting curve was performed during SYBR Green qRT-PCR analyses. For evaluation of the PCR amplification efficiency, a mix of all PCR-ready cDNA samples was used as a standard for the generation of a standard curve from serial dilutions. All efficiency values were 90% or higher, and all correlation coefficients (R^2^ values) were above 0.99.

The primers used for PCR and the corresponding sources [26,60,61,62,63,64,65,66] are listed in Appendix A.

### 4.6. Immunofluorescence

For immunofluorescence analysis, cells grown on coverslips were first washed with PBS to wash off residues of the cell medium and cell stimulants used. Cells were then fixed by a 30 min treatment in 4% PFA at RT, washed twice in PBS, and then permeabilized for 5 min in 0.1% Triton X100/PBS, or fixation and permeabilization was carried out by a 10 min incubation with 100% ice-cold methanol. This was followed by a wash step with PBS and incubation in 2% BSA/PBS solution for 15 min to block non-specific binding of antibodies. The cells were then incubated in fresh 2% BSA/PBS incubation solution with an antibody against CTNNB (GeneTex, Irvine, CA, USA, cat. no. GTX101435; dilution 1:100) or hTERT (Rockland, Limerick, PA, USA, cat. no. 600-401-252S) for 1 h. Subsequently, the coverslips were washed four times in PBS, and another 15 min blocking step was performed. This was followed by incubation in a freshly prepared second antibody (goat anti-rabbit Alexa Fluor 488; dilution 1:500) in incubation solution for 1 h in the dark. The specificity of the fluorescent signal was confirmed by immunofluorescence staining without a primary antibody. Subsequently, the coverslips were washed four times in PBS and once in ddH_2_O and dried. The coverslips were embedded in ProLong^TM^ gold antifade reagent (Invitrogen Carlsbad, CA, USA).

### 4.7. Co-Culture of HIBCPP Cells with HCPEnCs and iHCPEnCs

Co-culture of HIBCPP cells with HCPEnCs or iHCPEnCs (p20, p50, p100) on cell culture filter inserts (Greiner Bio One, Frickenhausen, Germany, cat. no. 662641) and measurement of TEER was performed as described previously [21].

## 5. Conclusions

In this study, we investigated whether immortalized endothelial cells of the human CP (iHCPEnCs) stably represent the CP endothelium. We demonstrate a high concordance of iHCPEnCs with their primary counterpart (HCPEnCs) on the transcriptome level. Although various components of the endothelial barrier-relevant Wnt signaling were detected in HCPEnCs as well as iHCPEnCs, the activation of Wnt signaling was only marginal, indicating the downregulation of CTNNB signaling in the CP endothelium. Importantly, in functional comparisons with HCPEnCs, iHCPEnCs up to passage 100 retained their contribution to barrier function at the CP. In conclusion, iHCPEnCs are a suitable in vitro model of the CP endothelium over extended passages.

## Figures and Tables

**Figure 1 ijms-26-01779-f001:**
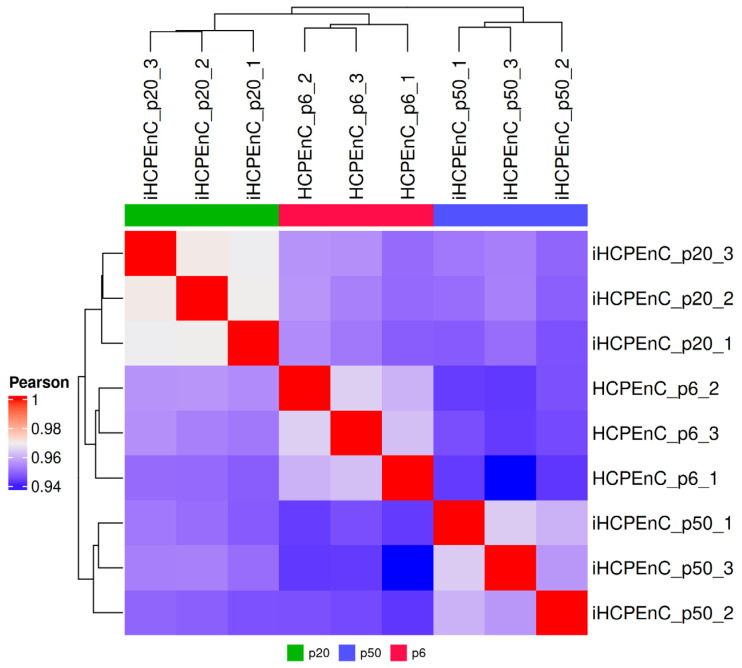
Graphical comparison of the transcriptomes of HCPEnCs and iHCPEnCs. The heat map illustrates the three biological replicates (*n* = 3) of HCPEnCs p6, iHCPEnCs p20, and iHCPEnCs p50 and depicts the respective levels of relationship among them. The three cell lines were juxtaposed in a matrix, their similarity was determined using hierarchical clustering, and they are displayed in the form of dendrograms. In addition, the calculated clusters are colored according to their Pearson coefficients, with red corresponding to a value of 1 and, thus, the highest possible match, while blue was assigned to the lowest identified value of 0.94.

**Figure 2 ijms-26-01779-f002:**
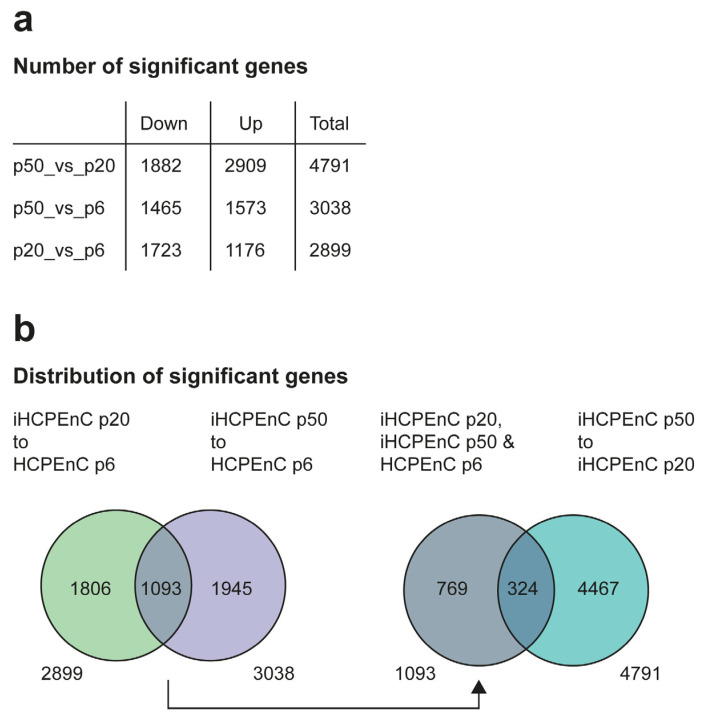
Comparison of the transcriptomes between HCPEnCs p6, iHCPEnCs p20, and iHCPEnCs p50. MACE was performed in triplicate (*n* = 3). (**a**) To determine significant genes, the transcriptome comparisons of p50_vs_p20, p50_vs_p6, and p20_vs_p6 were performed and divided into Down, Up, and Total. The values listed in (**a**) were used to compare and contrast them schematically in (**b**). The significant genes from p20_vs_p6 and p50_vs_p6 were compared with each other and then split into genes that occur in both groups or individually. The group of the significant genes occurring in both groups determined in this way was then compared with the significant genes from p50_vs_p20.

**Figure 3 ijms-26-01779-f003:**
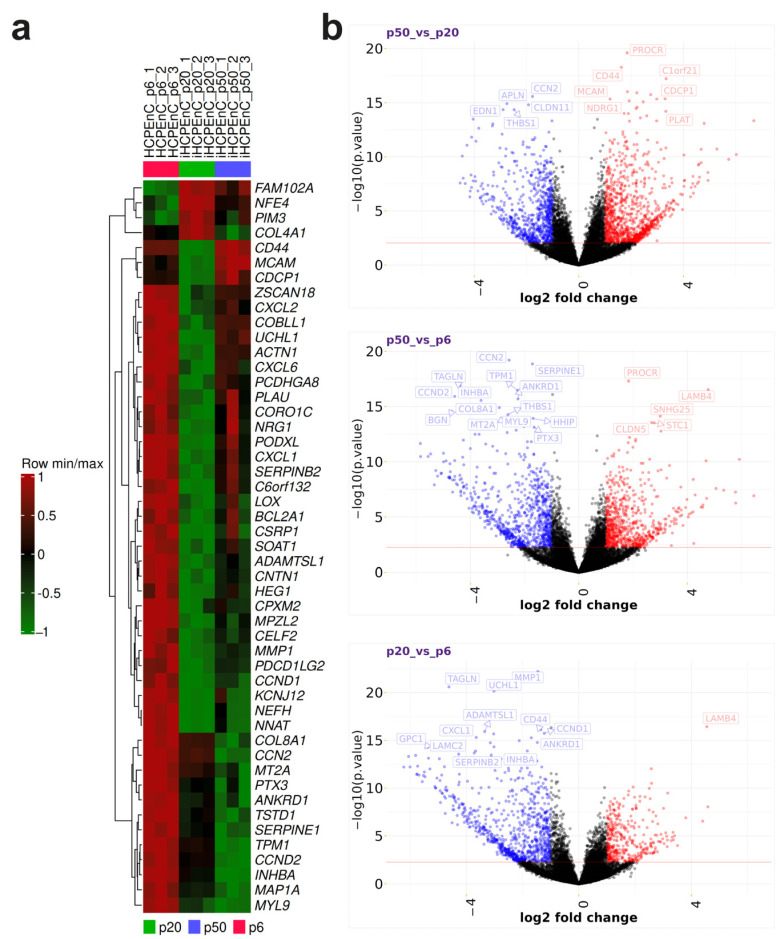
(**a**) The top 50 significantly altered genes in a heat map. MACE was performed in triplicate (*n* = 3). The gene expression data were standardized (i.e., converted into the corresponding Z-scores). Strong deviations of gene expression from the expected value (the mean of the detected expression levels of a gene) are assigned according to a scale from 1 to −1, with the value 1 as a strong positive gene expression (red) and −1 as a strong negative gene expression (green). Positive values can be regarded as upregulated and negative values as downregulated, while the value 0 corresponds to no deviation and, thus, represents no difference in expression levels. The resulting series minima and series maxima were then hierarchically clustered based on their similarity, allowing easy identification of the major differences between the primary cells, p6, and the immortalized cells, p20 and p50. (**b**) Contrasting individual volcano plots of p20_vs_p6, p50_vs_p6, and_p50_vs_p20. The log2 fold changes of each transcript were plotted against the negative log10 of their *p*-values for p50_vs_p20, p50_vs_p6, and p20_vs_p6. Here, genes with log2 fold changes below −1 were considered significantly underexpressed, and those above 1 were considered significantly overexpressed. In addition, a *p*-value of 2.5 was chosen as the significance threshold. The resulting significantly underexpressed and overexpressed genes are marked in blue and red, respectively, while the non-significant ones are marked in black.

**Figure 4 ijms-26-01779-f004:**
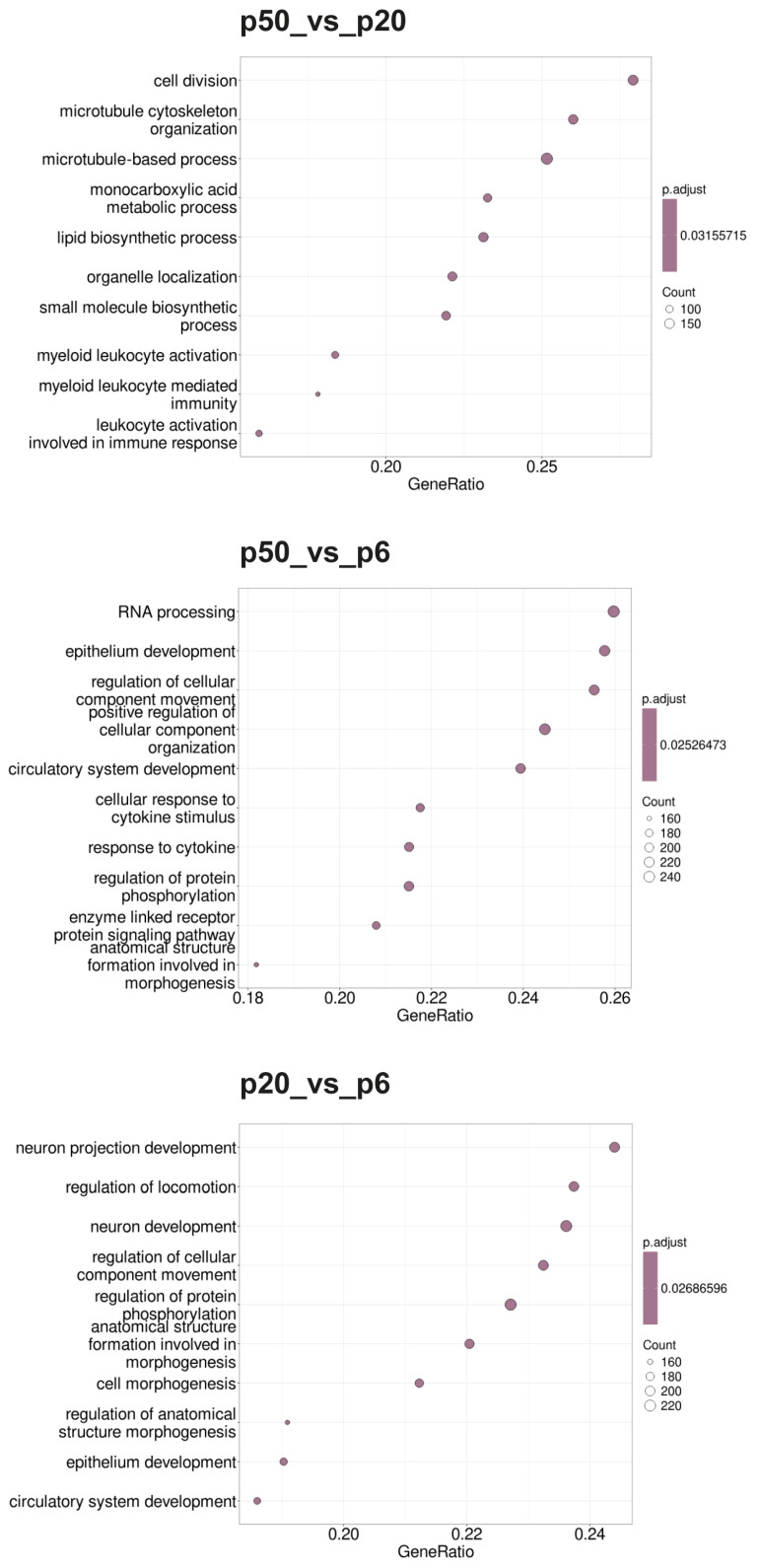
Dot blots of the GSEA-determined GOBPs. The GO database under the BP heading was used as the basis for the GSEA. The 10 GSs with the highest GeneRatios and lowest adjusted *p*-values were selected. After adjusting the *p*-values, these sets had the same *p*-values and were thus plotted according to the GeneRatios. The respective numbers (count) of significantly altered nuclear genes of the GSs are symbolized by circles (dots) with different diameters. The GeneRatio represents the ratio between counts per set size. MACE was performed in triplicate (*n* = 3).

**Figure 5 ijms-26-01779-f005:**
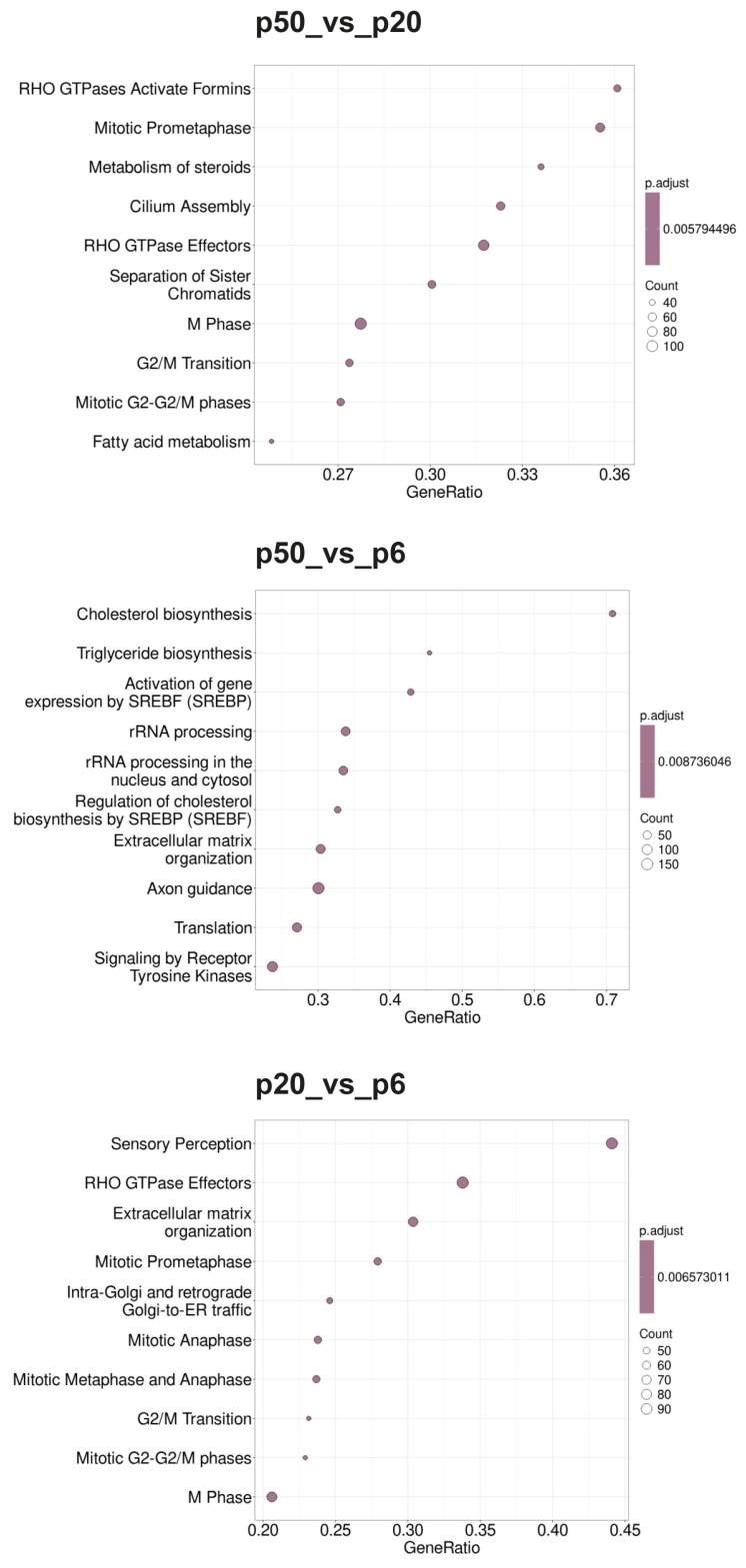
Dot blots of the Reactome pathways determined using the GSEA. The Reactome pathway database was used as the basis of the GSEA. The 10 GSs with the highest GeneRatios and lowest adjusted *p*-values were selected. After adjusting the *p*-values, these sets had the same *p*-values and were thus plotted according to the GeneRatios. The respective numbers (count) of significantly altered nuclear genes of the GSs are symbolized by circles (dots) with different diameters. The GeneRatio represents the ratio between counts per set size. MACE was performed in triplicate (*n* = 3).

**Figure 6 ijms-26-01779-f006:**
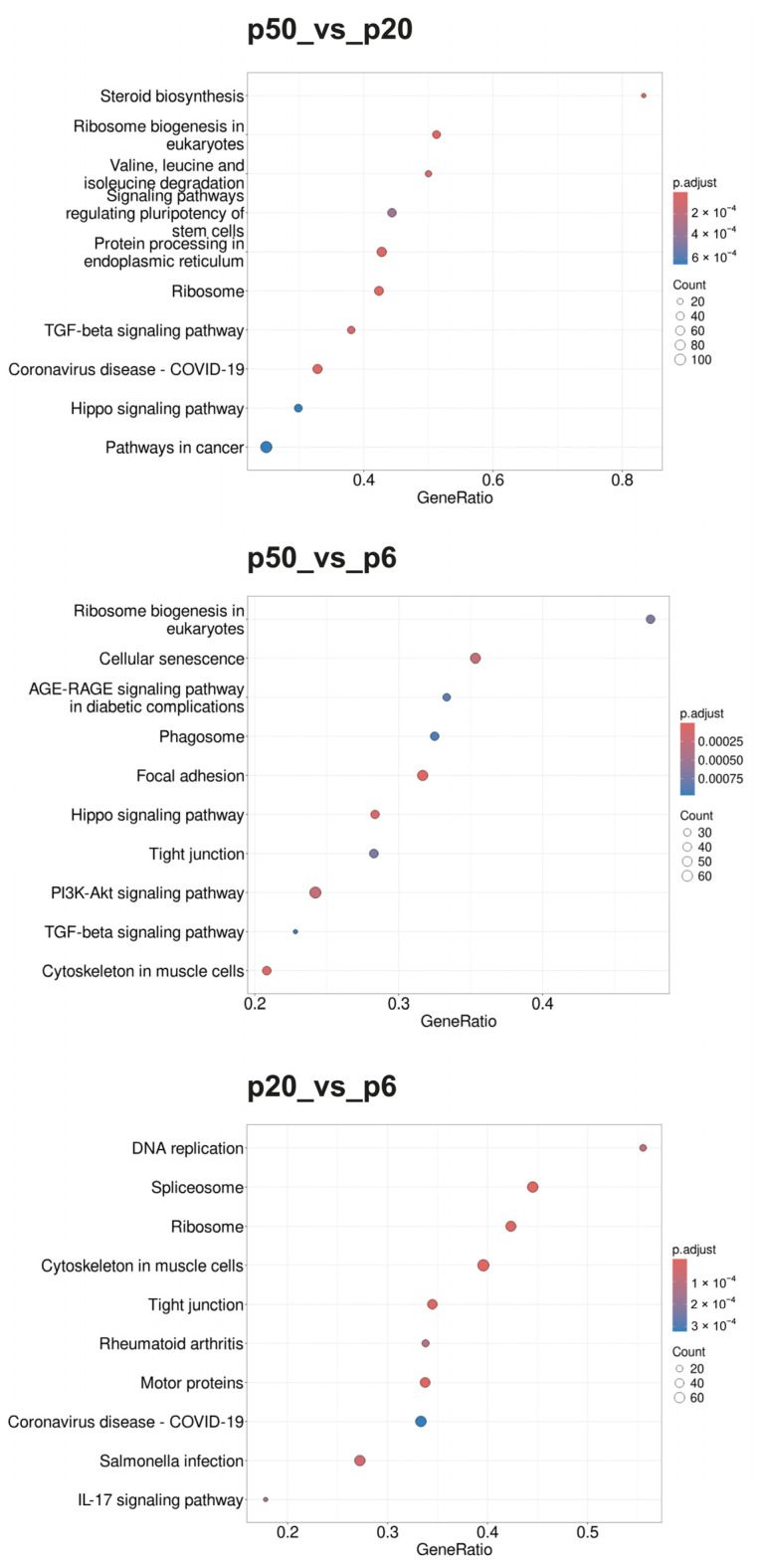
Dot blots of the KEGG pathways determined using the GSEA. The KEGG pathway database was used as the basis of the GSEA. The 10 GSs with the highest GeneRatios and lowest adjusted *p*-values were selected. After adjusting the *p*-values, these sets had the same *p*-values and were thus plotted according to the GeneRatios. The respective numbers (count) of significantly altered nuclear genes of the GSs are symbolized by circles (dots) with different diameters. The GeneRatio represents the ratio between counts per set size. MACE was performed in triplicate (*n* = 3).

**Figure 7 ijms-26-01779-f007:**
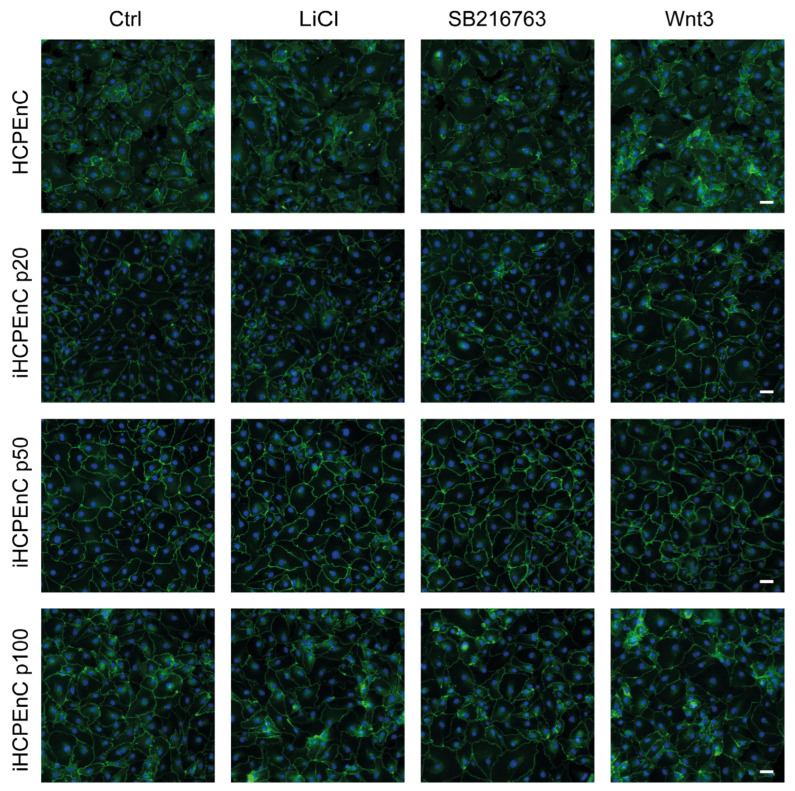
Immunofluorescence analysis comparing the distribution of CTNNB in HCPEnCs, iHCPEnCs p20, iHCPEnCs p50, and iHCPEnCs p100 “mock” treated (Ctrl) or treated with 10 mM LiCl, 10 µM SB216763, and 200 ng/mL Wnt3a. The staining shows no or only marginal nuclear accumulation of CTNNB (green) in HCPEnCs and iHCPEnCs treated for 24 h. Scale bar, 50 µm. Data show a representative result of at least four independent experiments (*n* = 4).

**Figure 8 ijms-26-01779-f008:**
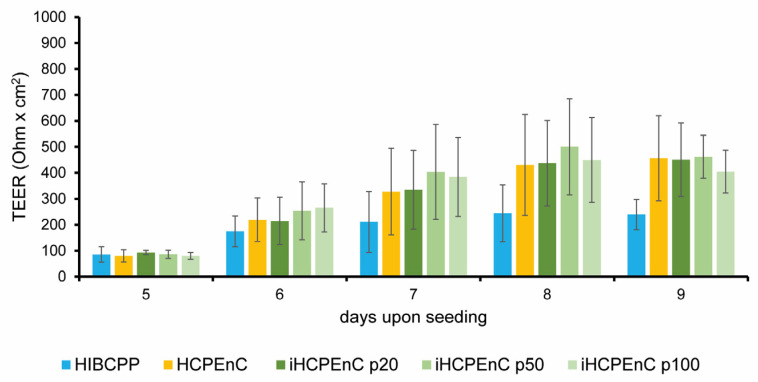
Functional analysis of a two-cell-type in vitro model composed of HIBCPP cells in co-culture with HCPEnCs, iHCPEnCs p20, iHCPEnCs p50, or iHCPEnCs p100. HIBCPP cells were grown in an inverted culture model, and HCPEnCs or iHCPEnCs, respectively, were seeded on the upper side of the filter membrane on day 5 [21]. The TEER developed by HIBCPP cells co-cultured over 4 days (until day 9) with HCPEnCs or iHCPEnCs, respectively, was compared to the TEER developed by HIBCPP cells cultured in the absence of HCPEnCs or iHCPEnCs. Three independent experiments (*n* = 3) were performed in quadruplicates. Data are represented as mean ± SD.

**Table 1 ijms-26-01779-t001:** Quantitative PCR for validation of transcriptome data from MACE. The relative fold changes determined using the efficiency-corrected 2^−ΔΔCT^ method are compared with the DEGs and the FDR values and *p*-values of the MACE. In both cases, each sample was based on three replicates, from which a mean value with an associated standard deviation (Stdev) or a normalized value was determined. Fold change or differential expression was determined in relation to the primary HCPEnCs.

Gene	iHCPEnC	qRT-PCR	Stdev	MACE	FDR	*p*-Value
*VWF*	p20	6.45	0.19	2.58	6.70 × 10^−10^	9.32 × 10^−13^
	p50	0.83	0.75	−0.71	0.13	0.02
*PLAT*	p20	0.08	0.01	−3.11	4.41 × 10^−11^	3.48 × 10^−14^
	p50	1.69	0.42	0.22	0.57	0.23
*THBS1*	p20	1.29	0.04	−0.14	0.40	0.19
	p50	0.24	0.16	−2.61	1.08 × 10^−11^	5.29 × 10^−15^
*SERPINE1*	p20	0.58	0.02	−0.98	2.44 × 10^−13^	4.58 × 10^−17^
	p50	0.30	0.08	−1.71	1.83 × 10^−15^	1.38 × 10^−19^
*EDN1*	p20	1.29	0.06	0.25	0.18	0.04
	p50	0.14	0.09	−2.64	2.43 × 10^−10^	2.20 × 10^−13^

## Data Availability

All MACE data described in this study have been submitted to Gene Expression Omnibus (http://www.ncbi.nlm.nih.gov/projects/geo/; accessed on 1 October 2023) under accession number GSE244600.

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
