# Peer review of "Transcriptome and Functional Comparison of Primary and Immortalized Endothelial Cells of the Human Choroid Plexus at the Blood–Cerebrospinal Fluid Barrier"

_ijms, 2025, doi:10.3390/ijms26041779_

Round 1
Reviewer 1 Report (Previous Reviewer 1)
Comments and Suggestions for Authors
Denzer and fellows conducted an experiment to investigate the primary and immortalized endothelial cells of the human choroid plexus. The manuscript is totally better than before, while some of the statement need to be revised then I think it could be accepted.
1. Line 569-570 R is a coding language, bioconductor is a R package, Rstudio is a platform to use R. The belonging relationship is confused.
Author Response
Denzer and fellows conducted an experiment to investigate the primary and immortalized endothelial cells of the human choroid plexus. The manuscript is totally better than before, while some of the statement need to be revised then I think it could be accepted.
We thank the reviewer for the positive statement.
- Line 569-570 R is a coding language, bioconductor is a R package, Rstudio is a platform to use R. The belonging relationship is confused.
We thank the reviewer for this comment. To express the mentioned statement more clearly, we now write in the Materials and Methods:
“R (version 4.4.2), a widely used programming language, was mainly used for statistics and data analysis. Most of the functions and tools used in this analysis were obtained from Bioconductor (version 3.20), a public source with a collection of programme libraries for the processing of biological data. Rstudio (version 2025.04.0) was used to edit the source code.”
Reviewer 2 Report (Previous Reviewer 3)
Comments and Suggestions for Authors
The authors present an interesting article in which they have examined various characteristics and biological traits of endothelial cell cultures of differing biological nature with respect to their ‘passaging’ age. It is well established that primary cells in particular are prone to ‘biological drift’ the longer they are in culture, and in particular primary brain endothelial cells are sensitive to such. As research in very much dependent on consistency in terms of models employed this study represents an important examination of how primary and immortalised counterparts of brain endothelial cells perform in this context. Briefly, the authors profiled the various cultures with respect to various genes and their expression/levels in the cells at different passages/in response agonists of Wnt signalling, and ultimately demonstrated that extended passaging of the aforementioned immortalised cells are comparable to the primary derived counterparts.
In reviewing this manuscript for an additional time I made couple of small observations, some of which originate in my original reviews and could be improved upon. The following should be considered when preparing a suitable revision.
1. While the immunofluorescence data in Figure 6 is good as a representative image, could the authors have performed colocalization analyses or similar to fully quantify the amount of CTNNB in the nucleus?
2. Some supplementary information is given to validate the primers used however it is not mentioned whether the primers used were validated fully to ensure they comply with MIQE guidelines. Information on how this was examined should be given and whether or not the primers performed to within acceptable performance parameters should be highlighted.
Author Response
The authors present an interesting article in which they have examined various characteristics and biological traits of endothelial cell cultures of differing biological nature with respect to their ‘passaging’ age. It is well established that primary cells in particular are prone to ‘biological drift’ the longer they are in culture, and in particular primary brain endothelial cells are sensitive to such. As research in very much dependent on consistency in terms of models employed this study represents an important examination of how primary and immortalised counterparts of brain endothelial cells perform in this context. Briefly, the authors profiled the various cultures with respect to various genes and their expression/levels in the cells at different passages/in response agonists of Wnt signalling, and ultimately demonstrated that extended passaging of the aforementioned immortalised cells are comparable to the primary derived counterparts.
We thank the reviewer for stating that our “study represents an important examination”.
In reviewing this manuscript for an additional time I made couple of small observations, some of which originate in my original reviews and could be improved upon. The following should be considered when preparing a suitable revision.
- While the immunofluorescence data in Figure 6 is good as a representative image, could the authors have performed colocalization analyses or similar to fully quantify the amount of CTNNB in the nucleus?
We agree with the reviewer that colocalization studies of CTNNB with further cellular markers are possible and would probably give a more quantitative impression of the amount of CTNNB in the nucleus. Still, from our experience, these colocalization studies are rather difficult and do not necessarily result in a reliable quantification. We are of the opinion that those analyses are beyond the scope of the present manuscript, but have included a note in the Discussion that further required investigations includes “colocalization analyses with further cellular markers to obtain a more quantitative impression” (page16, lines 489-490).
- Some supplementary information is given to validate the primers used however it is not mentioned whether the primers used were validated fully to ensure they comply with MIQE guidelines. Information on how this was examined should be given and whether or not the primers performed to within acceptable performance parameters should be highlighted.
We have now validated the PCR primers as it is described in the Materials and Methods section of the revised manuscript:
“PCR primers for qRT-PCR were validated as follows, taking into account the MIQE guidelines (Bustin et al. 2009). To verify primer specificity and accuracy, PCR product size was confirmed by agarose gel electrophoresis, and a melting curve was performed during SYBR Green qRT-PCR analyses. For evaluation of PCR amplification efficiency, a mix of all PCR-ready cDNA samples was used as standard for generation of a standard curve from serial dilutions. All efficiency values were 90% or higher, and all correlation coefficients (R2 values) were above 0.99.”
During primer validation we decided to remove YWAHZ as reference gene due to an efficiency value below 90%. RPL13a and SDHA are used as reference genes in the revised version of the manuscript. Furthermore, we recalculated standard deviations due to an error made during the previous calculation. We apologize for our neglect. The recalculated values in Table 1 do not affect the conclusions drawn from the table or the overall conclusions and message of the manuscript.
We sincerely thank the reviewer for insisting on this point of criticism, which has helped to decisively increase the quality of the manuscript.
Reviewer 3 Report (Previous Reviewer 2)
Comments and Suggestions for Authors
The authors have incorporated amendments in accordance with the comments previously submitted.
Author Response
The authors have incorporated amendments in accordance with the comments previously submitted.
We thank the reviewer for the approval of our revised manuscript.
This manuscript is a resubmission of an earlier submission. The following is a list of the peer review reports and author responses from that submission.
Round 1
Reviewer 1 Report
Comments and Suggestions for Authors
Overall, the research on HCPEnC by Denzer and fellows was interesting. The experiment was well organized, the images of molecular experiment are convincial. I recommend to accept after major revision. Please see the comments below:
1. About the transcriptome data: only GSEA was conducted. What about other enriched analysis like KEGG/GO enrichment? You need to indicate one of them without any bargain.
2. You found and defined so many DEGs. How this DEGs played their role in pathway regulation? Please analysis and clarify.
3. The Discussion is a little lack of logic. You need to make the writing more logical and scientific.
4. The necessary section Conclusion is required. Please added.
Reviewer 2 Report
Comments and Suggestions for Authors
In the present study, the authors undertake a thorough investigation into the characterisation of primary human CP endothelial cells. To this end, they have obtained previously immortalised HCPEnC cells that have undergone multiple passages and subjected them to extensive cDNA end analysis. The study's novelty and the contributions it makes to the scientific field are noteworthy. The methods employed are appropriate, and the presentation of the results is both clear and unambiguous. The discussion systematically progresses the reader through the authors' line of reasoning, providing a cogent rationale that contextualizes their perspective within the broader framework of existing knowledge.
Specific comments:
Line 96 - The description of the research results obtained should be removed from the introduction.
Line 470 - 'Cell culture' should be used in place of 'tissue culture'.
Line 571 - The methodology employed to assess the specificity of primary and secondary antibodies should be delineated.
Reviewer 3 Report
Comments and Suggestions for Authors
Th authors present an interesting study in which the characteristics of endothelial cells are compared are various passages to determine the differences following repeated culturing. Given the loss of traits by primary brain endothelial cells, much of cerebrovascular research is dependent on reliable, consistent immortalised cell types, so a study of this nature is of utmost interest. Briefly, the authors perform an expansive screening of various genes in terms of their expression/levels in the cells at different passages/in response agonists of Wnt signalling, ultimately indicating that the extended passages of the cells are comparable to the primary derived counterparts.
In reviewing this manuscript I made couple of small observations. The following should be considered when preparing a suitable revision.
1. Overall the design of the figures is very good, however I feel the label in many instances could be improved upon in terms of size and formatting. In many instances the text is very hard to read. The authors should consider adjusting this in any resubmission.
2. While the immunofluorescence data in Figure 6 is good as a representative image, could the authors have performed colocalization analyses or similar to fully quantify the amount of CTNNB in the nucleus?
3. The n-number should be clearly stated in each figure legend.
4. Were the primers that were used validated to comply with MQIE guidelines?